# Activity in perirhinal and entorhinal cortex predicts perceived visual similarities among category exemplars with highest precision

Kayla M Ferko[1,2†], Anna Blumenthal[1,3†], Chris B Martin[4], Daria Proklova[1], Alexander N Minos[1], Lisa M Saksida[1,2,5], Timothy J Bussey[1,2,5], Ali R Khan[1,2,6,7], Stefan Köhler[1,8*]

[1]Brain and Mind Institute, University of Western Ontario, London, Canada; [2]Robarts Research Institute Schulich School of Medicine and Dentistry, University of Western Ontario, London, Canada; [3]Cervo Brain Research Center, University of Laval, Quebec, Canada; [4]Department of Psychology, Florida State University, Tallahassee, United States; [5]Department of Physiology and Pharmacology, University of Western Ontario, London, Canada; [6]School of Biomedical Engineering, University of Western Ontario, London, Canada; [7]Department of Medical Biophysics, University of Western Ontario, London, Canada; [8]Department of Psychology, University of Western Ontario, London, Canada

**\*For correspondence:**
stefank@uwo.ca

[†]These authors contributed equally to this work

**Competing interest:** The authors declare that no competing interests exist.

**Abstract** Vision neuroscience has made great strides in understanding the hierarchical organization of object representations along the ventral visual stream (VVS). How VVS representations capture fine-grained visual similarities between objects that observers subjectively perceive has received limited examination so far. In the current study, we addressed this question by focussing on perceived visual similarities among subordinate exemplars of real-world categories. We hypothesized that these perceived similarities are reflected with highest fidelity in neural activity patterns downstream from inferotemporal regions, namely in perirhinal (PrC) and anterolateral entorhinal cortex (alErC) in the medial temporal lobe. To address this issue with functional magnetic resonance imaging (fMRI), we administered a modified 1-back task that required discrimination between category exemplars as well as categorization. Further, we obtained observer-specific ratings of perceived visual similarities, which predicted behavioural discrimination performance during scanning. As anticipated, we found that activity patterns in PrC and alErC predicted the structure of perceived visual similarity relationships among category exemplars, including its observer-specific component, with higher precision than any other VVS region. Our findings provide new evidence that subjective aspects of object perception that rely on fine-grained visual differentiation are reflected with highest fidelity in the medial temporal lobe.

## Editor's evaluation

Your response has been thorough and thoughtful and we believe this work now represents an important advancement to our understanding of the contributions of anterior temporal lobe regions in visual representations. Your approach affords tremendous specificity in the conclusions one can draw about the relationship between visual similarity and neural similarity along this ventral visual pathway and highlights perirhinal cortex as a potential key region whose neural representational structure relates to subjective behavior.

## Introduction

The ability to perceive similarities and differences between objects plays an integral role in cognition and behaviour. Perceived similarities are important, for example, for categorizing a fruit at the grocery store as an apple rather than a pear. The appreciation of more fine-grained similarities between exemplars of a category also shapes behaviour, such as when comparing different apples in order to select one for purchase. Indeed, experimental work in psychology has confirmed that the similarity of objects influences performance in numerous behavioural contexts, including but not limited to categorization, object discrimination, recognition memory, and prediction (see *Medin et al., 1993*; *Goldstone and Son, 2012*; *Hebart et al., 2020*, for review). Yet, despite the well-established links to behaviour, how the brain represents these similarities between objects is only beginning to be understood. A central question that has received limited investigation so far is how fine-grained visual similarities that observers subjectively perceive among subordinate category exemplars map onto neural object representations. Answering this question can provide insight as to what brain regions provide the 'read-out' for such subjective reports. Moreover, this endeavor holds promise for understanding how differences in the way in which observers perceive their visual environment are reflected in variations in functional brain organization (*Charest and Kriegeskorte, 2015*).

Functional neuroimaging, combined with pattern analysis techniques, provides a powerful tool for examining the mapping between similarity relationships in visual perception of objects and similarities in corresponding neural representations (*Kriegeskorte and Kievit, 2013*). A significant body of research addressing this issue has focused on the characterization of category structure and other coarse object distinctions, such as animacy. Findings from this research indicate that activity patterns in a large expanse of the ventral visual stream (VVS), often referred to as inferotemporal (IT) cortex or ventral temporal cortex, capture much of this structure in the environment (*Kriegeskorte et al., 2008*; *Connolly et al., 2012*; *Mur et al., 2013*; *Proklova et al., 2016*; *Cichy et al., 2019*). For example, numerous studies have revealed a similarity-based clustering of response patterns for objects in IT that is tied to category membership (e.g., *Kriegeskorte et al., 2008*; *Proklova et al., 2016*; see *Grill-Spector and Weiner, 2014* for review). In these studies, and in most related work, the primary focus has been on similarity in relation to object distinctions that are defined in objective terms, and on the characterization of neural representations that is shared by observers. As such, they do not address whether activity patterns in IT also capture similarity relationships among objects that characterize subjective aspects of visual perception that may vary across individuals' reports. When neural activity that corresponds to subjectively perceived visual similarities has been examined, extant research has mostly focussed on specific object features, such as shape or size (*Op de Beeck et al., 2008*; *Haushofer et al., 2008*; *Schwarzkopf et al., 2011*; *Moutsiana et al., 2016*) rather than on similarities between complex real-world objects that differ from each other on multiple dimensions. A notable exception to this research trend is an fMRI study that focussed on perceived similarities among select real-world objects that are personally meaningful (e.g., images of observers' own car, their own bicycle; *Charest et al., 2014*), which demonstrated links between observer-specific perceived similarity and the similarity structure embedded in activity patterns in IT.

In order to reveal the mapping between fine-grained perceived visual similarities among category exemplars and similarities in neural activity, it may not be sufficient to focus on activity in IT but critical to consider VVS regions situated downstream on the medial surface of the temporal lobe. Perirhinal cortex (PrC), and the primary region to which it projects, that is, lateral entorhinal cortex, are of particular interest when such perceived similarity relationships concern entire objects. The representational–hierarchical model of object processing in the VVS asserts that structures in the medial temporal lobe constitute the apex of its processing hierarchy (*Murray and Bussey, 1999*; *Bussey and Saksida, 2007*; *Cowell et al., 2010*; *Kent et al., 2016*). It proposes that there is a progressive integration of object features along the VVS that allows for increasing differentiation in the representation of objects from posterior to more anterior regions in the service of perception as well as other cognitive domains (e.g., recognition memory). Visual objects are thought to be represented in PrC in their most highly integrated form based on complex feature conjunctions (*Murray and Bussey, 1999*; *Buckley and Gaffan, 2006*; *Bussey and Saksida, 2007*; *Graham et al., 2010*; *Kent et al., 2016*), which support perceptual discrimination of even highly similar exemplars from the same object category (e.g., *O'Neil et al., 2013*; *O'Neil et al., 2009*). Lateral entorhinal cortex (or its human homologue anterolateral entorhinal cortex [alErC; *Maass et al., 2015*; *Yeung et al., 2017*]) has been suggested to extend this role in visual

object discrimination through integration with additional spatial features (*Connor and Knierim, 2017*; *Yeung et al., 2017*). Taken together, these properties make PrC and alErC ideally suited for providing the read-out for subjective reports of perceived visual similarity among the subordinate exemplars of object categories. Although more posterior VVS regions may also predict perceived similarity, the representational–hierarchical model asserts that they do not provide the same level of differentiation as regions at the apex in the medial temporal lobe. Moreover, they may also not capture those aspects of perceived similarity structure that are observer specific.

There is some evidence from human lesion studies suggesting that the integrity of regions in the medial temporal lobe is critical for perceiving the similarity among complex objects when it is high and objects cannot be easily discriminated from each other. This evidence comes from experiments that used variants of the oddity-discrimination task. In this task, participants view multiple objects (a minimum of three) on a computer screen and are asked to report the item that is least similar to the others. A finding documented in multiple reports (*Buckley et al., 2001*; *Barense et al., 2007*; *Bartko et al., 2007*; *Inhoff et al., 2019*; cf., *Stark and Squire, 2000*; *Levy et al., 2005*; *Hales and Clark, 2015*) is that patients with medial temporal lobe lesions that include PrC and alErC show deficits in oddity-discrimination tasks when complex objects with substantial feature overlap are compared, but not when oddity judgements require comparison of simple visual stimuli that can be distinguished based on a single feature such as colour or luminance. While the results of this lesion research have critically informed theoretical arguments that PrC plays a role in perceptual discrimination of objects (see *Bonnen et al., 2021*, for a recent computationally focused review), it is important to note that they do not provide a characterization of similarity structure of neural representations in PrC and alErC, nor a characterization of the transformation of representations from more posterior VVS regions to these regions in the medial temporal lobe. Moreover, lesion findings do not speak to whether neural representations in the medial temporal lobe capture the perceived similarity structure that is unique to individual observers.

In the current fMRI study, we tested the idea that the visual similarity structure among exemplars of real-world categories that observers subjectively perceive is predicted with higher precision by the similarity structure of neural activity in PrC and alErC than in the more posterior VVS regions traditionally considered in neuroscience investigations of visual object perception, including IT. During scanning, we administered a novel experimental task that required visual discrimination between consecutively presented exemplars from real-world categories as well as categorization (see *Figure 1*). In addition, we obtained ratings of perceived visual similarities among these exemplars from each observer offline (see *Figure 2*), as well as estimates of similarity derived from an influential computational model that describes objects at their intermediate visual feature level (HMAX, *Riesenhuber and Poggio, 1999*; *Cadieu et al., 2007*; *Serre et al., 2007*; *Khaligh-Razavi and Kriegeskorte, 2014*). At the behavioural level, we found that discrimination performance was highly sensitive to fine-grained perceived similarity among exemplars, including those aspects that were observer specific. Representational similarity analyses (RSAs) of ultra-high-resolution fMRI data revealed, in line with our hypotheses, that activation patterns in PrC and alErC predict this fine-grained structure within categories with higher precision than any other VVS region, and that they predict even those aspects of similarity structure that are unique to individual observers.

## Results

### Perceived visual similarity structure among exemplars varies across observers

We used inverse multidimensional scaling (iMDS, *Kriegeskorte and Mur, 2012*) to create participant-specific models of perceived visual similarity for 4 exemplars from 10 different categories (see *Figure 1*). Specifically, participants were instructed to arrange images of objects in a circular arena by placing those they perceived to be more visually similar closer together, and those they perceived to be less visually similar farther apart. Participants completed these arrangements offline, that is, outside of the scanner, in two phases, with the first phase involving sorting of the full set of 40 objects in a single arrangement (*Figure 1—figure supplement 1– Figure 1A*). The second phase required sorting of exemplars within categories in 10 separate arrangements (one per category; *Figure 1A*). Given our interest in representations that capture fine-grained object similarities within categories, our

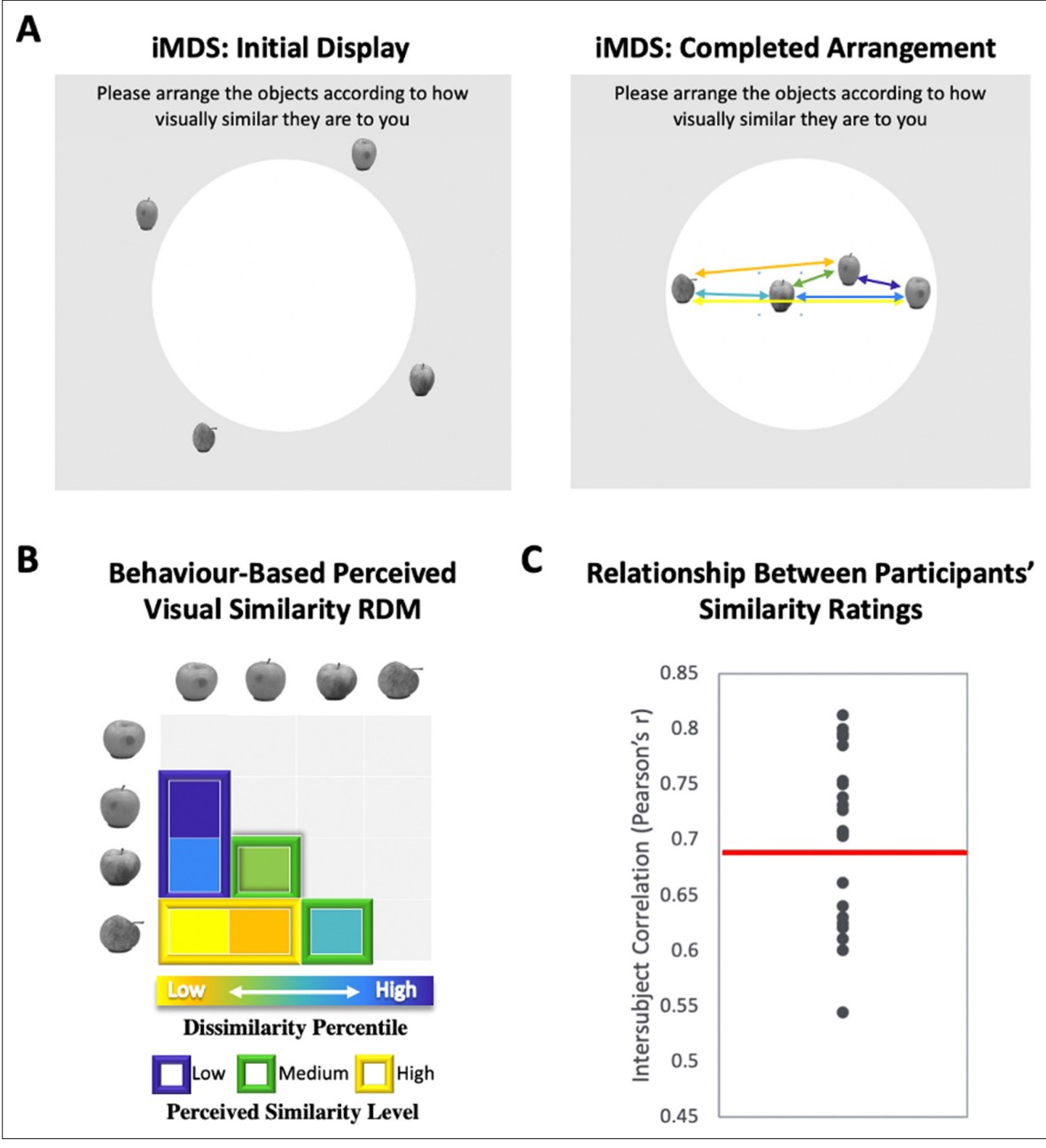

**Figure 1.** Perceived visual similarity ratings obtained offline with inverse multidimensional scaling (iMDS). (**A**) Task required placement of all exemplars from each category in circular arena, with distances reflecting perceived visual similarity. Arrows indicate the six pairwise distances used to compute representational dissimilarity matrix (RDM). (**B**) Behaviour-based RDM computed using dissimilarity (1 − Pearson's *r*) and conversion to percentiles for individual observers. Only values below diagonal were included. Six distances (between four exemplars) per category were rank ordered and grouped into three levels of similarity (low, middle, and high; for more detail, *Figure 1—figure supplement 1*). (**C**) Intersubject correlations for perceived similarity ratings across all exemplars and categories. Correlations were computed between each participant's RDM with the mean RDM (excluding the participant). Red horizontal line marks mean intersubject correlation, with variability in perceived visual similarity structure across observers reflected in the range displayed.

The online version of this article includes the following figure supplement(s) for figure 1:

**Figure supplement 1.** Stimuli employed and behavioural data obtained for specific object categories.

**Figure supplement 2.** In a follow-up study, a separate group of 30 participants completed 2 sessions of the inverse multidimensional scaling (iMDS) task for the 10 object categories separated by 7 ± 1 days.

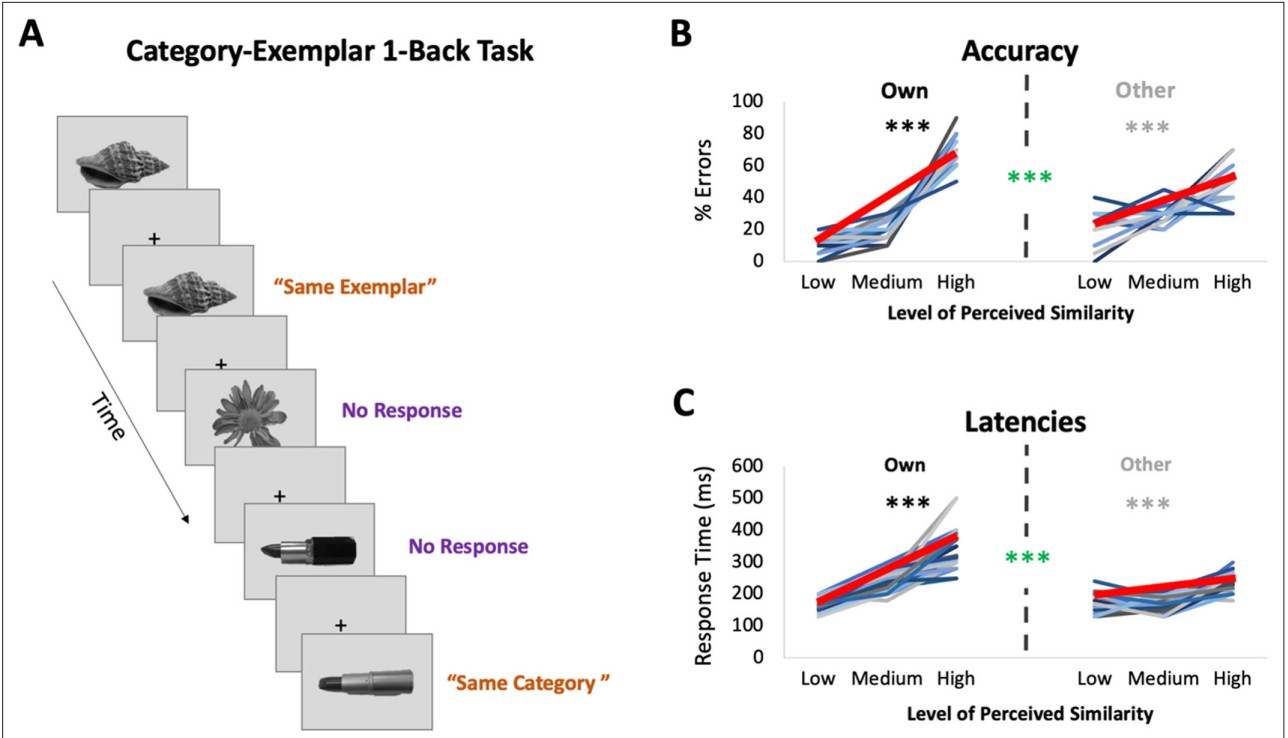

**Figure 2.** fMRI task: Category-Exemplar 1-Back Task. (**A**) Images of objects depicting one of the 4 exemplars from 10 different categories were presented. Participants indicated repetitions on catch trials with two different button presses depending on whether the image was an exact repeat of the one previously presented (same exemplar, same category) or a repeat at the category level (different exemplar, same category). The majority of trials (75%) reflected no repetitions on either level and required no response. Only trials that required no response (noncatch trials) were included in the fMRI analyses. (**B**) Percentage of errors that reflect incorrect same-exemplar responses on same-category trials as a function of perceived similarity (mean slope indicated with thick red line; for accuracy on all other trial types see ***Supplementary file 1***). Own values reflect behavioural performance as a function of participants' own visual similarity ratings; Other values reflect performance based on other participants' ratings (for a total of 22 iterations, which were then averaged). Error rate increased with increasing similarity as reflected in slopes (in black/grey ***p < 0.0001). Error rate was more sensitive to participants' own ratings as reflected in significantly higher slopes for the Own versus Other ratings (in green ***p < 0.0001). (**C**) Response times on correct same-category trials as a function of perceived similarity (mean slope indicated with thick red line). Own and Other values calculated as in (**B**). Response times increased with increasing similarity and were more sensitive to participants' own than other participants' ratings. Results in (**B, C**) show that behavioural performance on 1-back task is most sensitive to perceived similarity as reflected in participants' own ratings.

fMRI analyses relied on the similarity structures computed based on sorting in this second phase. The pairwise distances between all exemplars within each category were used to create a behaviour-based (i.e., subjective-report) representational dissimilarity matrix (RDM; ***Figure 1B***), which included a split of the range of similarities into three levels for sensitivity analyses in behaviour and neural activation patterns (see Methods for further detail). Examination of intersubject correlations of each participant's RDM and the mean of all other participants' RDMs (excluding their own) revealed a mean value of $r = 0.69$ ($t$-test $r > 0$: p < 0.001). When we calculated an RDM for ratings averaged across participants, and compared it with an RDM derived from a computational model developed to capture objects at their intermediate visual feature level (HMAX, ***Riesenhuber and Poggio, 1999***; ***Cadieu et al., 2007***; ***Serre et al., 2007***; ***Khaligh-Razavi and Kriegeskorte, 2014***), we also found a significant correlation ($r = 0.25$, p < 0.001), suggesting that average ratings capture a shared component in the subjective ratings that relate to objective image characteristics. Follow-up analyses that focussed on this relationship in different subranges of similarity among exemplars (low, medium, and high; see ***Figure 1B***) revealed, however, that the estimates derived from the HMAX model were only significantly correlated with average ratings at low and medium levels (low $r = 0.45$; p < 0.001; medium $r = 0.24$; p < 0.001; high $r = 0.05$; p > 0.05). This pattern suggests that at their finest grain, judgements of similarity within categories rely on integrated object representations that (1) are not captured by intermediate feature-level descriptions, and (2) are observer specific. Indeed, the range of intersubject correlations in reported similarity (calculated across exemplars and categories) that was present in our sample of participants

provides direct evidence for variability across observers ($r$ = 0.54–0.81; see *Figure 1C* and *Figure 1—figure supplement 1* for data on individual categories). There are also hints that this variability is most pronounced at the finest grain of judgements, as reflected in reduced intersubject correlations for high similarity exemplars (high $r$ = 0.64; medium $r$ = 0.65, low $r$ = 0.72; high < low; $t(22)$ = 3.41; $p$ < 0.001). Critically, in a separate behavioural experiment conducted in another sample of participants with multiple assessments, we found that individual differences in perceived visual similarity structure among exemplars are temporally stable observer characteristics (see *Figure 1—figure supplement 2*). We leveraged this variability across observers in subsequent analyses for assessing the precision of mapping in our fMRI results, examining whether the structure of neural representations in PrC and alErC also predict the perceived similarity structure that is unique to individual observers.

## Behavioural discrimination performance during scanning is sensitive to observers' perceived visual similarity structure

Participants underwent ultra-high-resolution fMRI scanning while completing a novel Category-Exemplar 1-Back Task designed to tax fine-grained visual object discrimination (see *Figure 2A*). This task required responding to two different types of repetition, namely repetition of same exemplars or of different exemplars from the same category, across consecutive trials. Participants were asked to provide a button-press response when they noticed repetitions, with different buttons for each type of repetition. On all other trials, participants were not required to provide a response. Importantly, this task was designed to ensure that participants engaged in categorization, while also discriminating between exemplars within categories. Performance on the Category-Exemplar 1-Back Task was sensitive to perceived visual similarity between exemplars as reflected in observers' offline ratings and formalized in the behavioural RDMs with 3 different levels of similarity (*Figure 2B,C*; see also *Supplementary file 1*). Specifically, response errors on same category trials increased with increasing visual similarity (significant linear slope; $t(22)$ = 18.35, $p$ < 0.0001). Moreover, response times for correct responses on same category trials were positively correlated with perceived visual similarity level ($t(22)$ = 13.47, $p$ < 0.0001). Critically, task performance was also sensitive to the unique perceived similarity structure within categories expressed by observers. When we compared the influence of participant's own similarity ratings with that of others on behaviour (*Figure 2B,C*), we found a significantly larger positive slope in error rate ($t(22)$ = 8.30; $p$ < 0.0001) and in response times ($t(22)$ = 9.68; $p$ < 0.0001) for participants' own ratings. This pattern of behaviour suggests that perceived visual similarity between exemplars influenced participants' discrimination performance during scanning, and, that it did so in an observer-specific manner.

## Patterns in multiple VVS regions predict perceived visual similarity structure among exemplars

To investigate whether the similarities between activation patterns in PrC and downstream alErC predict perceived similarities between exemplars in observers' reports, we employed anatomically defined regions of interest (ROIs) and created participant-specific models of neural similarity among category exemplars on the no-response trials. In order to examine the anatomical specificity of our findings, we also created such models for ROIs in other VVS and medial temporal lobe regions. Specifically, these ROIs included early visual cortex (EVC), lateral occipital cortex (LOC), posteromedial entorhinal cortex (pmErC), parahippocampal cortex (PhC), and temporal pole (TP) for comparison (see Figure 5 for visualization; note that LOC and PhC have typically been included in ROI definitions of IT in prior work; e.g., *Charest et al., 2014*; for direct comparison of results in IT and LOC, see *Figure 7—figure supplement 1*). Pairwise dissimilarities of neural patterns were employed to compute the brain-based RDMs (*Figure 3A,B*); Pearson's correlations were calculated so as to examine whether these RDM's predicted participants' own behaviour-based RDMs that were derived from their offline reports of perceived similarity (*Figure 3C*). Our analyses revealed that neural activation patterns in PrC and alErC did indeed correlate with participants' perceived visual similarity RDMs (Bonferroni-corrected $p$ < 0.01). Patterns in other regions of the VVS (EVC, $p$ < 0.003; LOC, $p$ < 0.002) were also significantly correlated with these behaviour-based RDMs. Critically, patterns in regions previously implicated in scene processing, specifically PhC and pmErC (*Schultz et al., 2015*; *Maass et al., 2015*; *Schröder et al., 2015*), did not predict the perceived similarity structure for objects (all $p$ > 0.5). Having established that activity patterns in multiple VVS regions predict perceived visual similarities

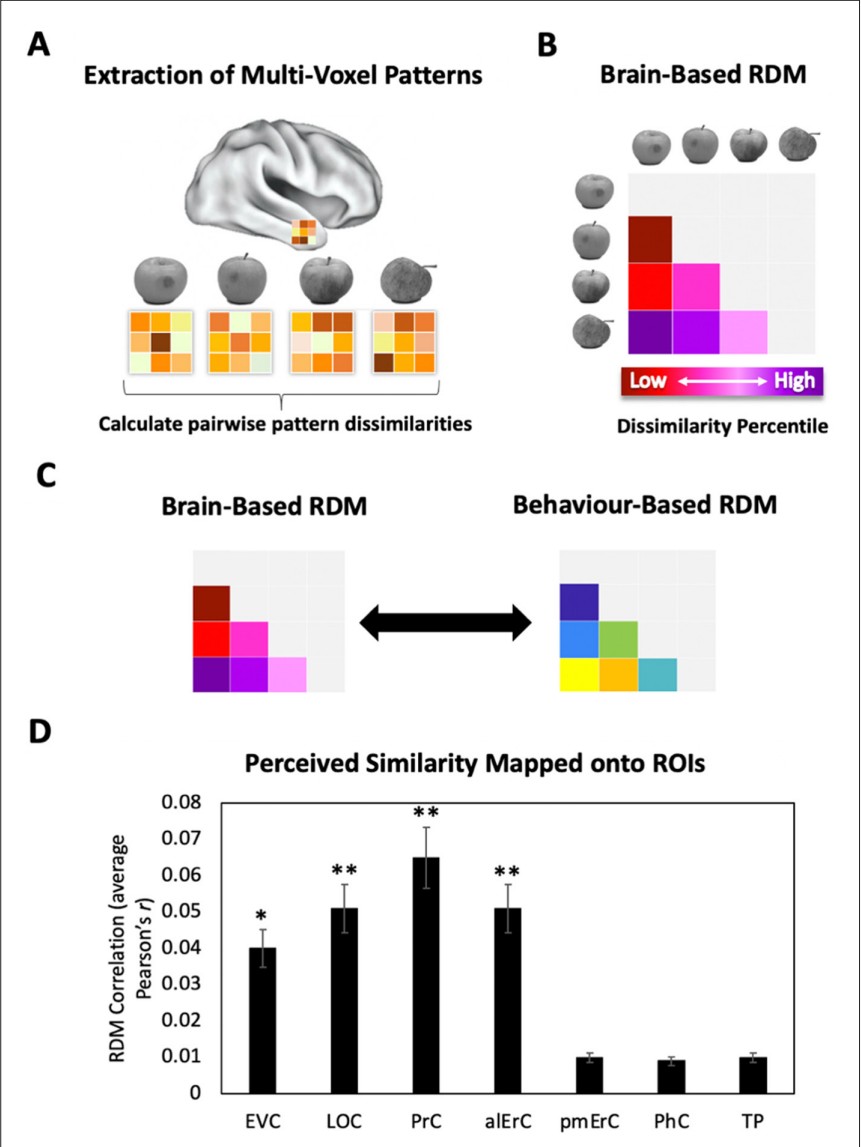

**Figure 3.** Brain-based representational dissimilarity matrices (RDMs) and their relationship to perceived visual similarity. (**A**) In each region of interest (ROI), mean multivoxel activation patterns were calculated for every exemplar using the no-response trials in the Category-Exemplar 1-Back Task. Pairwise pattern dissimilarities were computed as 1 − Pearson's *r*. (**B**) Pairwise pattern dissimilarity percentiles were used to create participant-specific brain-based RDMs. (**C**) Brain-based RDMs were correlated with participants' own behaviour-based similarity RDMs derived from their offline reports (black double arrows = within subject *r*). (**D**) Activation patterns in EVC, LOC, PrC, and alErC show significant correlations with participants' own perceived similarity ratings of objects (brain-based RDM × behaviour-based perceived similarity RDM within subjects; *p < 0.05, **p < 0.01 Bonferroni-corrected based on regions; error bars represent SEM). EVC = early visual cortex; LOC = lateral occipital complex; PrC = perirhinal cortex; alErC = anterolateral entorhinal cortex; pmErC = posteromedial entorhinal cortex; PhC = parahippocampal cortex; TP = temporal pole; see *Figure 5A* for visualization.

The online version of this article includes the following figure supplement(s) for figure 3:

**Figure supplement 1.** Temporal signal-to-noise ratio in perirhinal cortex (PrC; *M* = 12.25, SD = 2.68) and anterolateral entorhinal cortex (alErC; *M* = 10.80, SD = 3.05) in each of the 25 participants (average denoted by bolded red dot).

between exemplars, we followed up on this finding by asking whether PrC and alErC capture these similarities with higher fidelity than earlier regions (EVC and LOC). Towards this end, we next examined whether these regions predict similarity structure even when exemplars only differ from each other in subtle ways.

## PrC and alErC are the only regions whose patterns predict perceived visual similarity structure among exemplars when similarity is high

In this set of analyses, we examined the mapping between perceived similarity structure and neural activity patterns at a more fine-grained level within categories. Given prior evidence that PrC allows for the disambiguation of highly similar objects (*Murray and Bussey, 1999*; *Buckley and Gaffan, 2006*; *Bussey and Saksida, 2007*; *Graham et al., 2010*; *Kent et al., 2016*), we anticipated that patterns in PrC, and possibly downstream alErC, would represent even the most subtle visual differences that observers perceive among exemplars, whereas earlier VVS regions would not. To address this issue with RSA in our stimulus set, participants' behaviour RDMs, which were based on six pairwise distances between exemplars in each category, were divided into the three levels of similarity (low, medium, and high; see *Figure 1B* and *Figure 1—figure supplement 1* for range comparison). Recall that our behavioural analyses revealed, as previously described, that discrimination performance on the Category-Exemplar 1-Back Task was highly sensitive to these different levels of similarity. *Figure 4* displays the results of our level-specific fMRI analyses, which were conducted for those regions whose activity patterns showed significant correlations with participants' full perceived visual similarity space between exemplars (as shown in *Figure 3D*). Correlations between perceived similarity structure and activity patterns in PrC and alErC were significant at all three levels of similarity (Bonferroni-corrected $p < 0.01$; *Figure 4A*). In contrast, correlations for activity patterns in posterior VVS regions were significant only at the lowest level of perceived similarity (Bonferroni-corrected $p < 0.01$). This pattern of results cannot be attributed to differences in stability of activity patterns across levels of similarity for different regions; supplementary analyses revealed that stability was significant for all regions and did not interact with level of similarity (see *Figure 4—figure supplement 1*). Direct comparison between regions also revealed that correlations in PrC and alErC were higher than in LOC and EVC at medium and high levels of perceived similarity (Bonferroni-corrected $p < 0.05$; *Figure 4A*).

The described results suggest that object representations in posterior VVS regions may not have sufficient fidelity to allow for differentiation of exemplars that were perceived to be highly similar by observers, and that were most difficult to discriminate on our Category-Exemplar 1-Back Task. We followed up on this idea with complementary classification analyses of our fMRI data using a linear support vector machine (*Mur et al., 2009*). These analyses were conducted so as to examine in which VVS regions activity patterns associated with exemplars of high perceived similarity would be sufficiently separable so as to allow for classification as distinct items. They confirmed that activity patterns associated with specific exemplars can indeed be successfully classified in PrC and alErC at higher levels of perceived similarity than in posterior VVS regions (*Figure 4B* for further detail).

The analyses presented so far only focused on specific regions of interest. To answer the question of whether PrC and alErC are the only regions whose activity patterns predict perceived visual similarity among exemplars at its highest level, we also conducted whole-volume searchlight-based RSA. As expected based on our ROI analyses, patterns in PrC and alErC, as well as in earlier VVS regions, showed a predictive relationship in these searchlight analyses when the full range of perceived visual similarity values among exemplars was considered; no regions in the scanned brain volume outside of the VVS exhibited this predictive relationship (threshold-free cluster enhancement [TFCE] corrected $p < 0.05$; *Figure 5C,D*). Critically, our searchlight analyses revealed that PrC and alErC were indeed the only regions in the entire scanned brain volume whose patterns correlated with observer's reports when similarity between exemplars was perceived to be high, and objects were most difficult to discriminate on the Category-Exemplar 1-Back Task (TFCE-corrected $p < 0.05$; *Figure 5E,F*).

## Patterns in posterior VVS regions predict similarity structure that is shared by observers and tied to object characteristics at the intermediate feature level

Because our behavioural results point to shared as well as observer-specific components in the perceived similarity structure that participants reported, we conducted several additional sets of

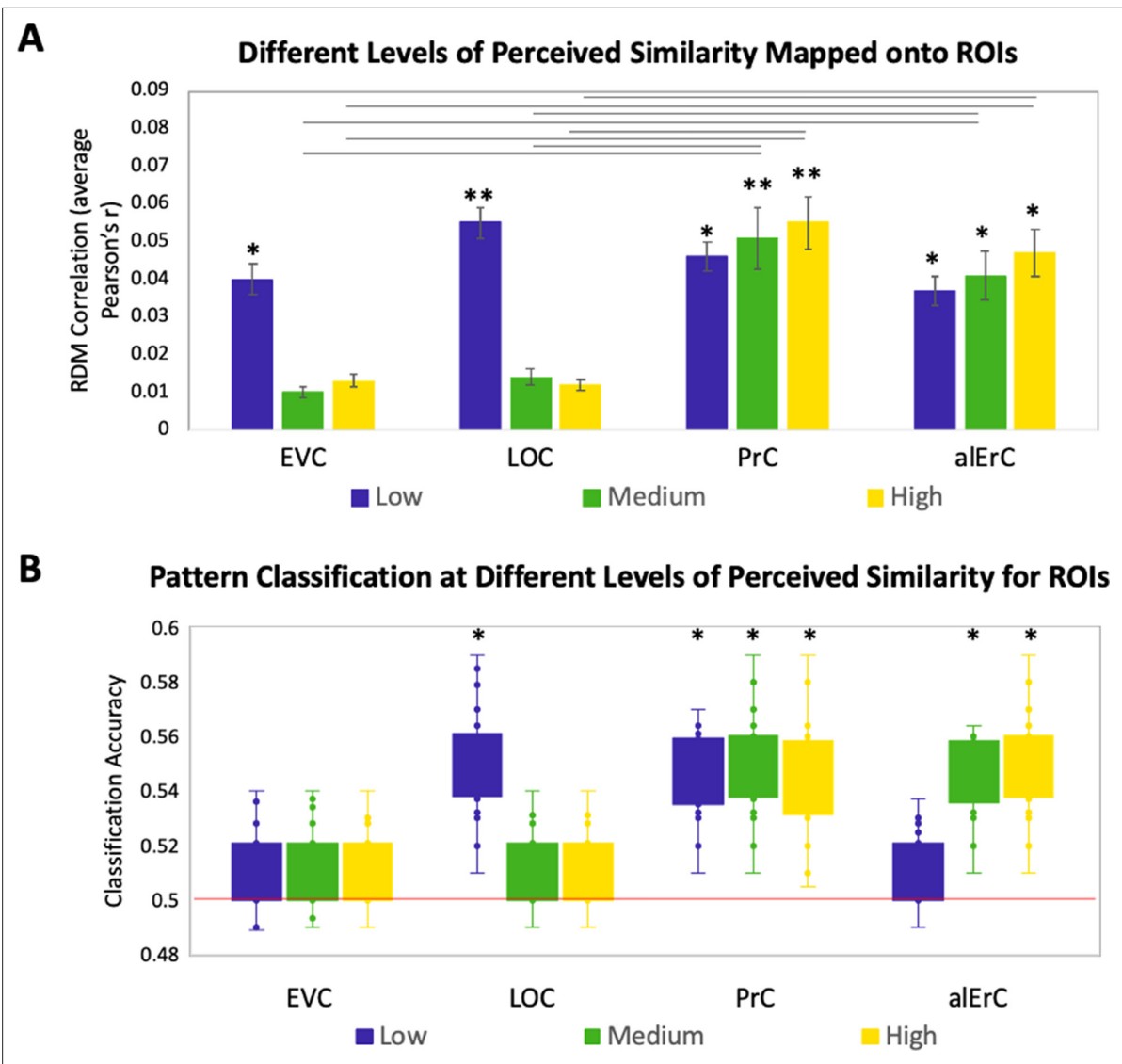

**Figure 4.** Relationship between brain representational dissimilarity matrices (RDMs) and reports at different levels of perceived visual similarity for region of interests (ROIs) showing significant effects in *Figure 3D*. (**A**) Correlation of brain-based RDM and participants' own behaviour-based RDM at low, medium, and high levels of similarity. Only activation patterns perirhinal cortex (PrC) and anterolateral entorhinal cortex (alErC) show significant correlation with ratings at middle and high levels of perceived similarity (**$p < 0.01$, *$p < 0.05$, Bonferroni-corrected for regions and levels). Correlations in PrC and alErC were significantly larger than those in early visual cortex (EVC) and lateral occipital cortex (LOC) at the medium and high levels of perceived similarity (horizontal lines indicate $p < 0.05$). (**B**) Box and whisker plots for classification accuracy of neural activation patterns at each level of perceived similarity in different ROIs. We adopted a common classification approach using linear support vector classifier and leave-one-run-out cross-validation (*Misaki et al., 2010*). Results from one tailed *t*-tests to probe whether classifier performance was above chance indicate that patterns in LOC are distinguishable only at lowest level of perceived similarity within categories. PrC and alErC are only regions in which patterns are distinguishable at medium and high levels of perceived similarity (*$p < 0.05$, Bonferroni-corrected).

The online version of this article includes the following figure supplement(s) for figure 4:

**Figure supplement 1.** Relationship between brain-based representational dissimilarity matrices (RDMs) in even and odd runs, within and between participants at different levels of perceived similarity.

analyses, aiming to address which VVS regions would show activity patterns that might predict these different components. In this context, we considered that the perceived similarity structure among exemplars that is shared across observers for low and medium levels is tied to objective image characteristics at the intermediate feature level, as reflected in the significant correlation between averaged similarity ratings and estimates derived from the HMAX model in our data (see Behavioural results). Given the assertion of the representational hierarchical model that objects are represented in PrC and

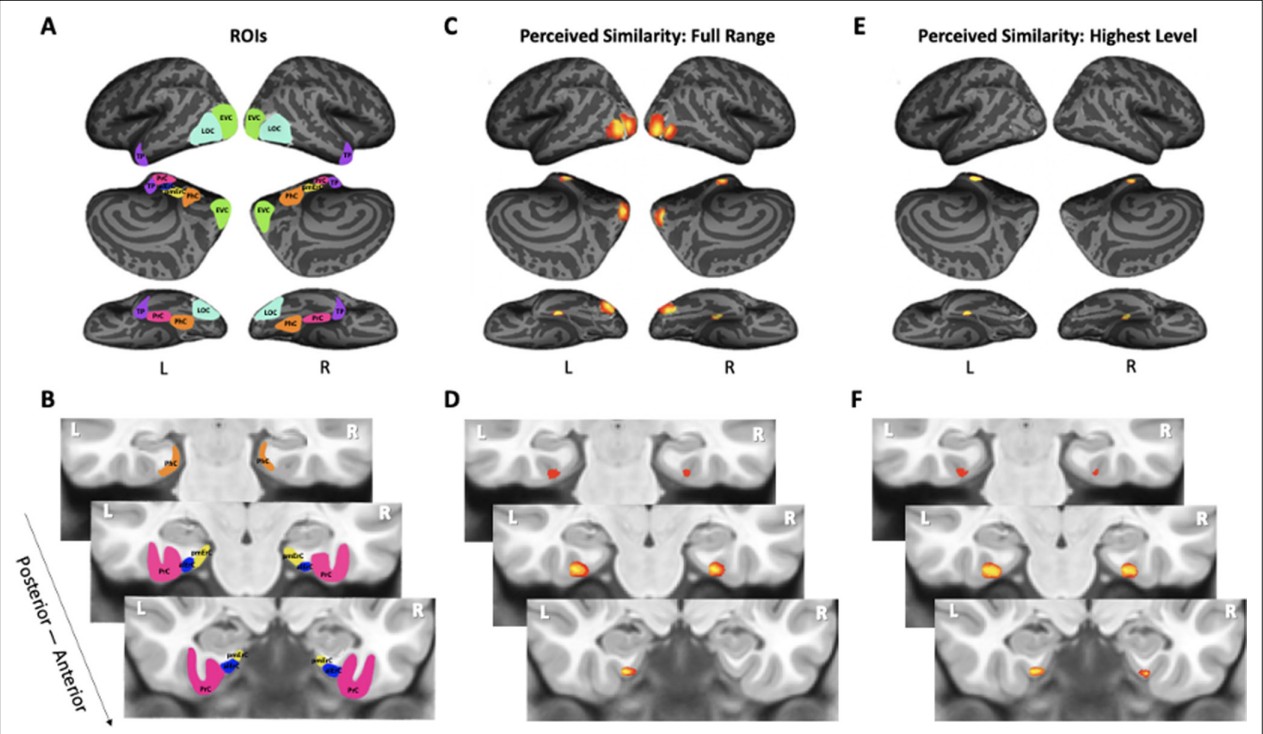

**Figure 5.** Visualization of region of interests (ROIs) and results from whole-volume searchlight analyses. (**A, B**) Visual depiction of ROIs. Early visual cortex (EVC = green), lateral occipital complex (LOC = cyan), perirhinal cortex (PrC = pink), parahippocampal cortex (PhC = orange), anterolateral entorhinal cortex (alErC = blue), posteromedial entorhinal cortex (pmErC = yellow), temporal pole (TP = purple). (**C, D**) Cortical regions revealed with searchlight analysis in which brain-based representational dissimilarity matrices (RDMs) were significantly correlated with behaviour-based perceived similarity RDMs across full range. Significant correlations were observed in PrC, alErC, and more posterior ventral visual stream (VVS) regions. (**E, F**) Cortical regions revealed with searchlight analysis in which brain-based RDMs were significantly correlated with behavioural RDMs at highest level of perceived similarity. Significant correlations were observed only in PrC and alErC. Maps are displayed with corrected statistical threshold of p < 0.05 at cluster level (using threshold-free cluster enhancement).

alErC based on complex feature conjunctions (rather than intermediate features), we predicted that the shared component of perceived similarity estimates of the HMAX model would predict activity patterns only in VVS regions posterior to PrC and alErC. Indeed, we found a significant predictive relationship of activity patterns in regions EVC and LOC (Bonferroni-corrected p < 0.01) but not in PrC and alErC (all Bonferroni-corrected p > 0.05; see *Figure 6A,B*) for average perceived similarity. This significant relationship in posterior VVS regions was also only present at low (for EVC and LOC) and medium levels of similarity (for LOC; see *Figure 6B*). Analyses that directly employed the estimates of similarity between object images obtained from the computational HMAX model (*Riesenhuber and Poggio, 1999*; *Cadieu et al., 2007*; *Serre et al., 2007*; *Khaligh-Razavi and Kriegeskorte, 2014*) revealed strikingly similar findings (see *Figure 6B vs D*). Activity patterns in EVC and LOC showed a significant correlation with the HMAX model at low and medium but not at the highest level of similarity; Bonferroni-corrected p < 0.01. Critically, activity patterns in PrC and alErC did not correlate with estimates from the HMAX model nor with average ratings at any level of object similarity (all Bonferroni-corrected p > 0.05; see *Figure 6C,D*). Together, these results suggest that activity in VVS regions posterior to PrC and alErC capture the components of perceived visual similarity structure among exemplars that is shared by observers and that is closely related to object features at the intermediate feature level. At the same time, these neural representations in posterior VVS regions do not appear to allow for differentiation of exemplars at high levels of perceived similarity that tend to be observer specific.

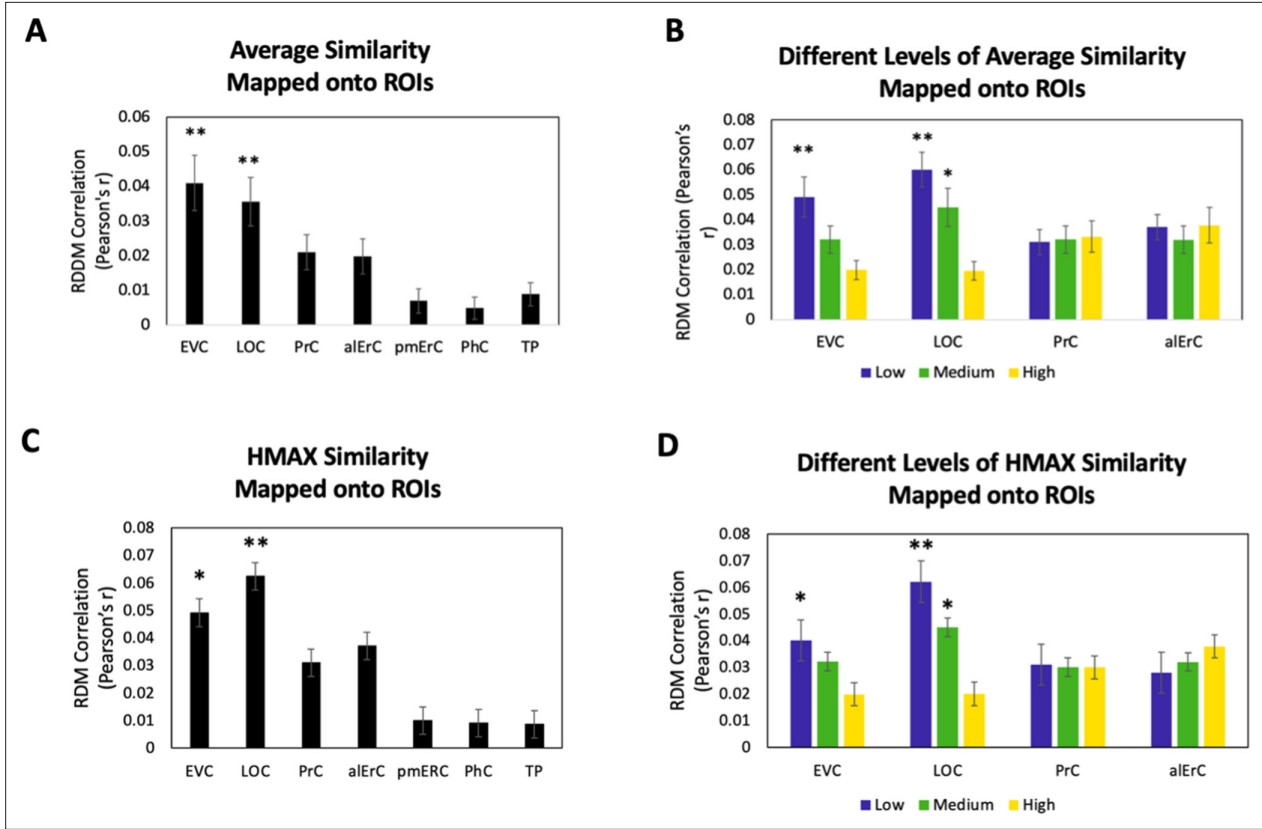

**Figure 6.** Brain-based representational dissimilarity matrices (RDMs) and their relationship to average perceived visual similarity. Brain-based RDMs were correlated with (**A**) the average behaviour-based similarity RDMs and (**B**) the different levels of average similarity RDMs; and with (**C**) the entire RDM derived from the HMAX model (**D**) the RDMs at different levels of similarity derived from the HMAX model. Patterns in EVC and LOC show relationship to average whole perceived similarity ratings **p < 0.01, Bonferroni-corrected based on regions. EVC and LOC also show correlations to the average low, and LOC to the medium level of perceived similarity *p < 0.05, Bonferroni-corrected. EVC = early visual cortex; LOC = lateral occipital complex; PrC = perirhinal cortex; alErC = anterolateral entorhinal cortex; pmErC = posteromedial entorhinal cortex; PhC = parahippocampal cortex; TP = temporal pole.

## Patterns in PrC and alErC predict fine-grained perceived visual similarity structure among exemplars in an observer-specific manner

In the final set of analyses, we directly focussed on the variability across participants' reports of similarity in order to address whether activity patterns in PrC and alErC predict even those perceived similarities with high precision that are unique to individual observers. We reasoned that if neural patterns in a region represent the observer's *unique* perceived similarity structure, brain–behaviour correlations should be higher when calculated within rather than between participants (*Figure 7A*, black versus grey arrows; see *Supplementary file 2* for similar results revealed with a multiple regression approach). In other words, if there are observer-specific relationships, activity patterns should predict participants' own perceived similarity structure better than someone else's. Such analyses would reveal that inter-individual differences are not only present in the perceptual reports of observers and in their discrimination performance, as demonstrated in our behavioural analyses, but also in corresponding neural representations in PrC and alErC. Indeed, initial analyses of our fMRI data demonstrated the presence of stable observer-specific activity patterns for the object exemplars probed in our study in all regions of interest (see *Figure 7—figure supplement 2*). The i-index introduced by *Charest et al., 2014*, which directly measures differences in correlations for the same (i.e., own) versus other observers, allowed us to examine which of these observer-specific activation patterns predict observer-specific structure in reports of perceived similarities. These analyses confirmed our expectation that the neural activation patterns in PrC and alErC predict observer-specific perceived visual similarity structure (Bonferroni-corrected p[within-participant $r$ > between-participant $r$] < 0.01). In contrast, activation patterns in posterior VVS regions, EVC and LOC (Bonferroni-corrected p[within-participant $r$ >

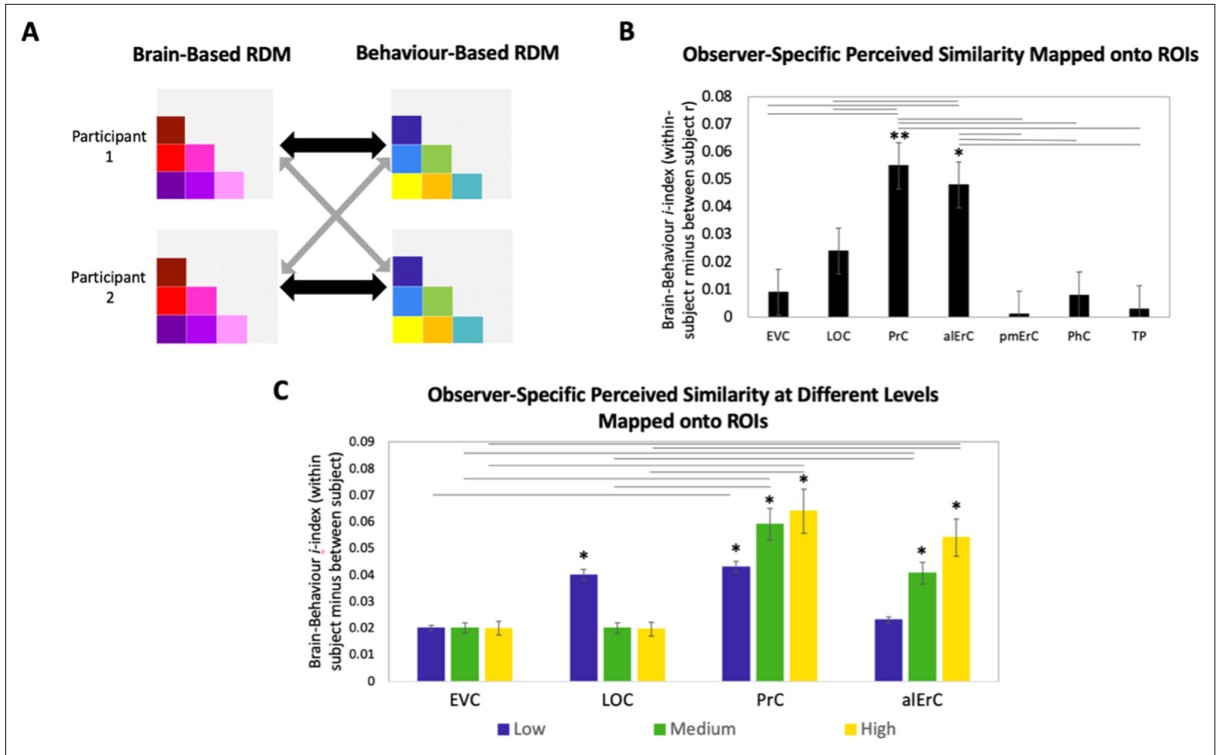

**Figure 7.** Brain-based representational dissimilarity matrices (RDMs) and their relationship to observer-specific perceived visual similarity. (**A**) Brain-based RDMs were correlated with (1) participants' own behaviour-based similarity RDMs (black double arrows = within subject *r*) and (2) other participants' behaviour-based similarity RDMs (grey arrows = between subject *r*) for comparison. (**B**) Patterns in PrC and alErC show relationship to perceived similarity ratings that are observer specific as reflected in brain–behaviour *i*-index (i.e., within minus between subject correlation; *p < 0.05, **p < 0.01, Bonferroni-corrected based on regions, with testing against a null distribution created by randomizing subject labels; error bars represent standard error of the mean [SEM] estimated based on randomization). PrC and alErC also show significant higher *i*-index than other regions as indicated with horizontal lines; *p < 0.05, Bonferroni-corrected. (**C**) Only patterns in PrC and alErC show relationship observer-specific perceived similarity ratings at the middle and high levels of perceived similarity; (*p < 0.05, Bonferroni-corrected for regions and levels). Correlations in PrC and alErC were significantly larger than those in EVC and LOC at the medium and high levels of perceived similarity (horizontal lines indicate p < 0.05). EVC = early visual cortex; LOC = lateral occipital complex; PrC = perirhinal cortex; alErC = anterolateral entorhinal cortex; pmErC = posteromedial entorhinal cortex; PhC = parahippocampal cortex; TP = temporal pole; see *Figure 5a* for visualization.

The online version of this article includes the following figure supplement(s) for figure 7:

**Figure supplement 1.** Comparison between inferotemporal (IT) cortex and lateral occipital complex (LOC): brain-based representational dissimilarity matrices (RDMs) and their relationship to perceived visual similarity (**A**) at all levels of perceived similarity (**p < 0.01, *p < 0.05) and (**B**) corresponding *i*-index; and (**C**) at different levels of perceived similarity and (**D**) corresponding *i*-index (*p < 0.05).

**Figure supplement 2.** Relationship between brain-based representational dissimilarity matrices (RDMs) in even and odd runs, within and between participants.

between-participant *r*] > 0.05) did not uniquely predict participants' own perceived similarity structure (*Figure 7B*). Not surprisingly, regions that did not predict participants' perceived similarity structure at all (pmErC, PhC, and TP; *Figure 3D*), also did not have significantly above zero *i*-indices (Bonferroni-corrected p[within-participant *r* > between-participant *r*] > 0.05). Critically, PrC and alErC showed significant brain–behaviour *i*-indices even when we restricted analyses to fine-grained differentiation, that is, to the subrange of high levels of perceived similarity (Bonferroni-corrected p[within-participant *r* > between-participant *r*] < 0.05). Taken together, these results reveal that activity patterns in PrC and alErC even predict perceived similarities that are unique to individual observers, which are most prevalent in fine-grained structure.

## Discussion

Vision neuroscience has made great strides in understanding the hierarchical organization of object representations along the VVS. How VVS representations capture fine-grained differences between objects that observers subjectively perceive has received limited examination so far. In the current study, we addressed this question by focussing on perceived similarities among exemplars of real-world categories. Using a novel Category-Exemplar 1-Back Task, we found that visual discrimination performance is highly sensitive to the visual similarity structure that is reflected in observers' subjective reports. Combining this task with fMRI scanning at ultra-high-resolution allowed us to show, in line with our general hypotheses, that activity patterns in PrC and alErC predict perceived visual similarities among exemplars with higher precision than any other VVS region, including prediction of those aspects of similarity structure that are unique to individual observers.

Research that has aimed to characterize the nature of object representations in human PrC with fMRI has shown that the degree of feature overlap between objects is captured by activation patterns in this region. Such a relationship has been revealed in multiple task contexts, with images of real-world objects and with words denoting such objects; moreover, it has been observed for feature overlap at the perceptual as well as the semantic level (*Clarke and Tyler, 2014*; *Erez et al., 2016*; *Bruffaerts et al., 2013*; *Martin et al., 2018a*). These findings, in combination with work from neurophysiology in nonhuman animals and from computational modelling, have been interpreted to suggest that PrC integrates features of objects with complex conjunctive coding into representations of whole objects, and that the resulting conjunctive representations allow for differentiation of objects even when they are highly similar due to a high degree of feature overlap (*Bussey et al., 2002*; *Murray and Bussey, 1999*; *Cowell et al., 2010*). Indeed, it is this type of conjunctive coding that has motivated the central notion of the representational–hierarchical model of VVS organization that PrC, together with alErC (*Connor and Knierim, 2017*; *Yeung et al., 2017*), can be considered the pinnacle of the VVS object-processing hierarchy. Using a metric that was rooted in participants' subjective reports of perceived visual similarity and that showed a direct relationship to behavioural discrimination performance, the current fMRI findings provide new support for this hierarchical model by revealing increased differentiation of subordinate category exemplars in PrC and alErC, as compared to more posterior VVS regions.

Findings from lesion studies conducted with oddity-discrimination tasks support the idea that medial temporal lobe structures downstream from IT play a critical role in processes required for the appreciation of fine-grained visual similarities between complex real-world objects that are expressed in perceptual reports (see *Bonnen et al., 2021*, for review). Numerous studies conducted in humans and in other species have shown that performance on such tasks relies on the integrity of PrC when objects with high visual feature overlap must be judged (*Buckley et al., 2001*; *Barense et al., 2007*; *Bartko et al., 2007*; *Inhoff et al., 2019*; cf., *Stark and Squire, 2000*; *Levy et al., 2005*; *Hales and Clark, 2015*). For example, *Barense et al., 2007* compared performance on multiple visual oddity tasks between individuals with lesions in the medial temporal lobe that largely spared PrC and ErC, versus individuals with more widespread damage in the medial temporal lobes that included PrC and ErC. Most notably, individuals in the latter but not in the former group showed impairments in identifying the odd-one out item in sets of images of real-world objects that shared a high number of overlapping visual features. While the results of these prior lesion studies are compatible with the conclusions we draw in the current study, they do not allow for characterization of similarity structure of neural representations in PrC and alErC, and their direct comparison with representations in other medial temporal and posterior VVS regions, as provided here.

The anatomical specificity of our fMRI findings in the medial temporal lobe is striking. While activity patterns that reflected the similarity structure among category exemplars were present in PrC and alErC, they were absent in medial temporal regions that have previously been implicated in visual discrimination of scenes, specifically pmErC and PhC cortex (see *Schultz et al., 2015*, for review). This specificity is noteworthy in light of documented differences in functional connectivity between these regions that have been linked to object versus scene processing, with PrC being connected to alErC, and PhC being connected to pmErC, respectively (*Maass et al., 2015*; see *Schröder et al., 2015* for more broadly distributed differences in functional connectivity between ErC subregions and other cortical structures).

The higher precision we observed for the representation of perceived similarity relationships in the medial temporal lobe, as compared to more posterior VVS regions, is of particular theoretical interest for the representational–hierarchical model of VVS organization (*Murray and Bussey, 1999*; *Bussey and Saksida, 2007*; *Cowell et al., 2010*; *Kent et al., 2016*). Most revealing, in this context, is the comparison between PrC and ErC versus LOC, a region that is part of the large swath of cortex that is often referred to as IT in neuroimaging research and that has been linked to processing of object shape in many prior fMRI studies (e.g., *Grill-Spector et al., 2001*; *Kriegeskorte et al., 2008*; *Connolly et al., 2012*; *Mur et al., 2013*; *Proklova et al., 2016*; *Cichy et al., 2019*). While activity patterns in LOC predicted some aspects of similarity structure among subordinate category exemplars in the current work, this relationship was observed at a coarser grain than in PrC and alErC; it only held when perceived visual similarity was low, and when performance in behavioural discrimination revealed that objects were easily distinguishable. Indeed, complementary pattern classification analyses revealed that activity patterns associated with exemplars of high perceived similarity were not sufficiently separable in LOC so as to allow for classification as distinct items. By contrast, this classification could be successfully performed based on activity patterns in PrC and alErC. Indeed, our searchlight analyses showed that these two regions in the medial temporal lobe were the only ones in which activity was related to perceived visual similarity at a fine-grained level.

Our analyses of the relationship between activity patterns and aspects of perceived similarity among exemplars that are tied to objective image characteristics, as estimated by the computational HMAX model, offer support for the central claim of the representational–hierarchical model that the transformation of object representations from IT (specifically, LOC) to medial temporal-lobe structures involves further integration. Notably, the component of perceived similarity structure that was shared by observers showed a statistical relationship to the estimates of the HMAX model, which describes objects at the intermediate feature level and which has been linked to LOC representations in prior work (*Riesenhuber and Poggio, 1999*; *Cadieu et al., 2007*; *Serre et al., 2007*; but see *Khaligh-Razavi and Kriegeskorte, 2014*, *Kubilius et al., 2016* for limitations as compared to deep convolutional neural network models). In the present study, we also found that HMAX estimates of the similarity among the exemplars we employed were correlated with activity patterns in LOC, but only at low and medium levels of similarity. The fine-grained similarity structure among exemplars that led to the largest number of confusion errors in behavioural discrimination on the Category-Exemplar 1-Back Task during scanning was predominantly observer specific, and this fine-grained observer-specific structure was solely predicted by activity patterns in PrC and alErC. The coding of objects in PrC and alErC as fully integrated entities based on complex feature conjunctions arguably affords the flexibility that is required to capture the fine-grained differentiation among exemplars that characterizes the perception of individual observers. .

Our study was not designed to directly address what factors might drive the variability in perceived similarity structure across observers that were present in their subjective reports, their discrimination performance, and in corresponding activity patterns in PrC and alErC. Prior evidence from fMRI research on neural representations in other VVS regions suggests that past experience and object familiarity may play an important role. *Charest et al., 2014* revealed observer-specific effects in the similarity structure of activity patterns in IT that were tied to participants' reports for highly familiar real-world objects with unique personal meaning (e.g., images of observers' own car, their own bicycle). Notably, this observer-specific mapping between similarity in activity patterns and reports was not present for unfamiliar objects. There is also evidence from behavioural training studies indicating that prior experience with categories has an impact on how the similarity among its exemplars is perceived. In a recent study by *Collins and Behrmann, 2020*, for example, it was shown that just a few days of repeated exposure can lead to increased differentiation among exemplars, and that these changes are most pronounced at the level of fine-grained similarity structure when observers have had some prior experience with the category in question. This change in similarity structure based on training occurred in the absence of any apparent opportunity to gain new sematic knowledge about the exemplars in question, suggesting it could reflect an increase in perceptual expertise. Indeed in recent behavioural research from our laboratory, we have found that the degree of self-reported exposure to real-world object categories, but not corresponding semantic knowledge, predicts observers' perceived visual similarity structure among exemplars, and that this relationship is most notable at the level of fine-grained similarity structure (*Minos et al., 2021*). It is possible that the observer specificity

in fine-grained differentiation among exemplars that was predicted by activity patterns in PrC and alErC in the current study is tied to similar factors at work; it may reflect interindividual differences in perceptual expertise across observers and categories. Although speculative at present, such an account would be in line with a large body of evidence revealing learning-related plasticity in object representations in these medial temporal lobe structures (see *Banks et al., 2014*, for review). This account can be directly tested with training paradigms that target specific real-world object categories in future fMRI research. Regardless of the outcome of such future research, the current findings highlight the critical value of probing the subjective appreciation of visual object similarities, and their variability across observers, for a complete understanding of the transformation of neural representations from posterior regions to those at the apex of the VVS.

## Materials and methods

### Participants

A total of 29 participants completed the perceived similarity iMDS arrangement task and fMRI experiment (12 females; age range = 18–35 years old; mean age = 24.2 years). All participants were right handed, fluent in English, and had no known history of psychiatric or neurological disorders. Three participants were removed due to excessive head motion above the cutoff of 0.8 mm of framewise displacement, one participant weas removed due to behavioural performance accuracy 2 SD below the average on the fMRI task, and two participants were removed due to poor signal quality in medial temporal-lobe regions, i.e., a temporal signal-to-noise ratio 2 SD below average (see *Figure 3—figure supplement 1 Figure 1*). Therefore, 23 participants were included in the final analyses. All participants gave informed consent, were debriefed, and received monetary compensation upon completion of the experiment. This study was conducted with Western's Human Research Ethics Board approval.

### Stimuli

In order to investigate object representations at the exemplar level, we selected stimuli with varying normative levels of perceived visual similarity from the Migo Normative Database (*Migo et al., 2013*). We used 40 greyscale images of objects from 10 categories (*Figure 1—figure supplement 1*). Each category in our set was made up of four exemplars that all shared the same name (e.g., apple, lipstick, and stapler). Based on findings from a pilot study (*n* = 40), only stimuli with perceived similarity ratings similar to those from the normative database were selected.

### Modelling of perceived visual similarity structure

In order to obtain observer-specific models of perceived visual similarity structure for our stimuli, participants provided reports of perceived similarity between all stimuli on a computer outside of the scanner prior to scanning. Participants were seated in front of a monitor and completed a modified version of the iMDS task (*Kriegeskorte and Mur, 2012*). Specifically, participants were asked to drag-and-drop images into a white circle (i.e., arena), and arrange them according to perceived visual similarity (*Kriegeskorte and Mur, 2012*; see *Figure 2A*). Objects perceived to be more visually similar were to be placed closer together, and objects perceived to be less visually similar were to be placed further apart. The iMDS task consisted of two phases. In the first phase, participants arranged all 40 stimuli according to perceived visual similarity. In the second phase, participants completed 10 category-specific trials in which they sorted 4 exemplars from the same category according to their perceived visual similarity. They were instructed to use the entire space within the circle, and make sure they compared each stimulus to every other stimulus. Only data from the second phase were considerd in the current analyses. A MATLAB-based toolbox was used to calculate distances between each pair of exemplars, and to convert these distances to dissimilarity percentiles (*Kriegeskorte and Mur, 2012*). These dissimilarity percentiles were then used to create observer-specific behaviour-based RDMs that represented each observers' perceived similarity space at the exemplar level. The behaviour-based RDMs for the entire range included 6 dissimilarity percentiles for each of the 10 categories (i.e., 6 × 10 = 60 dissimilarity percentiles).

These behaviour-based RDMs, which captured the full range of perceived similarity, were used to create three behaviour-based RDMs to reflect three levels of perceived similarity (low, medium, and high). The six pairwise distances (expressed as dissimilarity percentile) per category were sorted

into the two largest, two medium, and two smallest dissimilarities to create behaviour-based RDMs for low, medium, and high perceived visual similarity, respectively. These behaviour-based RDMs for each of the levels included 2 dissimilarity percentiles for each of the 10 categories (i.e., 2 × 10 = 20 dissimilarity percentiles). In order to ensure that the different levels of perceived similarity were nonoverlapping, any values that did not allow for at least 0.1 dissimilarity percentile between each of the successive levels (i.e., high-middle and middle-low) was excluded. The range of dissimilarity percentiles did not differ significantly between the different levels of perceived similarity (p > 0.05; *Figure 1—figure supplement 1C*).

## Modelling of objective visual similarity: HMAX model

We obtained estimates of similarities that were derived from a computational model, HMAX (*Riesenhuber and Poggio, 1999*; *Cadieu et al., 2007*; *Serre et al., 2007*; *Khaligh-Razavi and Kriegeskorte, 2014*), developed to describe objects at their intermediate visual feature level, including shape. In this biologically inspired model, simple cell layers, akin to V1 cells, detect local orientation using Gabor filters. These orientation signals are pooled in complex cell layers to extract global features. In this way, HMAX is designed as a four-layer hierarchical feed-forward structure similar to that previously described in the VVS. The output layer of HMAX captures an object's shape over activation patterns and has been shown to correspond to activation patterns in IT cortex and LOC (*Khaligh-Razavi and Kriegeskorte, 2014*). We used Matlab implementation of HMAX (https://maxlab.neuro.georgetown.edu/hmax.html) to extract the activations from the C2 layer of the model and compute RDM between 4 exemplars from the 10 categories used in the current study.

## Category-Exemplar 1-Back Task

For the main experiment, participants completed a variation of a 1-back task, coined the 'Category-Exemplar 1-Back' in the 3T scanner. We created this new 1-back task to ensure that participants were attending carefully to each individual object, given our interest in fine-grained object discrimination. As in a classic 1-back task, participants were shown a stream of individual objects and asked to indicate with a button press when the object was an exact repeat of the object previous to it, and no response was required when the object was from a different category as the one previous (*Figure 2A*). Our novel twist was the addition of a second response option, whereby participants were asked to indicate with a different button when the object was from the same category as the previous one, but a different exemplar. The two response trial types served as catch trials to ensure participants' attention focussed on differences between objects across consecutive trials, and to assess behavioural performance. These modifications of the classic 1-back task were introduced to ensure that participants considered category membership and engaged in object processing at the exemplar level. Successful identification of the repetition of different exemplars from the same category could not be based on local low-level features, such as changes in luminance, texture, or shape across consecutive trials. Participants used their right index and middle finger to respond, with response assignment counterbalanced across participants. Of the three trial types—same exemplar, same category, different category—only the no-response trials (i.e., different category) were used in the fMRI analysis to avoid motor confounds associated with button presses. By extension, none of the trials considered for assessment of similarity in activation patterns were immediate neighbours.

Participants completed a total of eight functional runs that each lasted 4 min (stimulus duration = 1.2 s, intertrial interval = 1 s). Run order was counterbalanced across participants. Within each run, each of the 40 exemplars were presented 3 times as no-response trials, and once as a catch trial, for a total of 24 presentations on no-response trials and 8 catch trials (excluded from fMRI analyses) per exemplar across the entire experiment. Prior to scanning, each participant completed a 5-min practice task with images from categories not included in the functional scanning experiment.

## fMRI data acquisition

MRI data were acquired using a 3T MR system (Siemens). A 32-channel head coil was used. Before the fMRI session, a whole head MP-RAGE volume (TE = 2.28 ms, TR = 2400 ms, TI = 1060 ms, resolution = 0.8 × 0.8 × 0.8 mm isometric) was acquired. This was followed by four fMRI runs, each with 300 volumes, which consisted of 42 T2*-weighted slices with a resolution of 1.7 × 1.7 mm (TE = 30 ms, TR = 1000 ms, slice thickness 1.7 mm, FOV 200 mm, parallel imaging with grappa factor 2). T2*-weighted

data were collected at this ultra-high resolution so as to optimize differentiation of BOLD signal in anterolateral versus posterior medial entorhinal cortex. The T2* slices were acquired in odd-even interleaved fashion in the anterior to posterior direction. Subsequently, a T2-weighted image (TE = 564 ms, TR = 3200 ms, resolution 0.8 × 0.8 × 0.8 mm isometric) was acquired. Finally, participants then completed four more fMRI runs. Total duration of MRI acquisition was approximately 60 min.

## Preprocessing and modelling

MRI data were converted to brain imaging data structure (*Gorgolewski et al., 2016*) and ran through fmriprep-v1.1.8 (*Esteban et al., 2019*). This preprocessing included motion correction, slice time correction, susceptibility distortion correction, registration from EPI to T1w image, and confounds estimated (e.g., tCompCor, aCompCor, and framewise displacement). Component based noise correction was performed using anatomical and temporal CompCor, aCompCor, and tCompCor, by adding these confound estimates as regressors in SPM12 during a first-level general linear model (GLM) (*Behzadi et al., 2007*). Each participant was coregistered to the participant-specific T1w image by fmriprep. First-level analyses were conducted in native space for each participant with no spatial smoothing to preserve ultra-high-resolution patterns of activity for multivariate pattern analyses (MVPA). Exemplar-specific multivoxel activity patterns were estimated in 40 separate general linear models using the mean activity of the no-response trials across runs.

## ROI definitions for fMRI analyses

Anatomical regions of interest were defined using multiple techniques. Automated segmentation was employed to delineate PrC, ErC, and PhC (ASHS; *Wisse et al., 2016*). We manually segmented each ERC obtained from ASHS into alErC and pmErC following a protocol developed by *Yeung et al., 2017*, which is derived from a functional connectivity study (*Maass et al., 2015*). A probabilistic atlas was used to define EVC (*Wang et al., 2015*) and TP (*Fischl, 2012*). A functional localizer was used to define LOC as the contiguous voxels located along the lateral extent of the occipital lobe that responded more strongly to intact objects than scrambled objects (p < 0.01, uncorrected; *Proklova et al., 2016*).

In the VVS, we focussed on lateral occipital complex and the TP as they have previously been linked to object processing (e.g., *Grill-Spector et al., 2001*; *Martin et al., 2018b*), as well as EVC. In the medial temporal lobe (MTL), we included ROIs for the posteromedial ErC and PhC, both of which have been linked to scene processing (e.g., *Maass et al., 2015*; *Schröder et al., 2015*; *Schultz et al., 2015*; *Epstein and Baker, 2019*).

## RSAs of fMRI data

For each ROI, linear correlation distances (Pearson's *r*) were calculated between all pairs of exemplar-specific multivoxel patterns using CoSMoMVPA toolbox in Matlab (*Oosterhof et al., 2016*) across all voxels. These correlations were used to create participant-specific brain-based RDMs (1 − Pearson's *r*), which capture the unique neural pattern dissimilarities between all exemplars within each category (*n* = 10), within each region (*n* = 8).

Whole-volume RSA were conducted using surface-based searchlight analysis (*Kriegeskorte et al., 2008*; *Oosterhof et al., 2016*; *Martin et al., 2018a*). Specifically, we defined a 100-voxel neighbourhood around each surface voxel, and computed a brain-based RDM within this region, analogous to the ROI-based RSA. This searchlight was swept across the entire cortical surface (*Kriegeskorte et al., 2008*; *Oosterhof et al., 2016*). First, the entire perceived similarity RDM for all within category ratings was compared to each searchlight. These brain–behaviour correlations were Fisher transformed and mapped to the centre of each searchlight for each participant separately. Participant-specific similarity maps were then standardized and group-level statistical analysis was performed. TFCE was used to correct for multiple comparisons with a cluster threshold of p < 0.05 (*Smith and Nichols, 2009*).

## Acknowledgements

This work was supported by a Canadian Institutes for Health Research Project Grant (CIHR Grant # 366062) to AK and SK, a Brain Canada Platform Support Grant, and the Canada First Research Excellence Fund. KF was funded through a Natural Sciences and Engineering Research Council doctoral Canadian Graduate Scholarship (NSERC CGS-D) and an Ontario Graduate Scholarship (OGS). AB was

funded through an Ontario Trillium Scholarship for Doctoral study in Canada. We thank Dr. Marieke Mur for her generous help with use of the iMDS task and related Matlab programs.

## Additional information

### Funding

| Funder | Grant reference number | Author |
|---|---|---|
| Canadian Institutes of Health Research | 366062 | Ali R Khan<br>Stefan Köhler |
| Natural Sciences and Engineering Research Council of Canada | | Kayla M Ferko |
| Ontario Trillium Foundation | | Anna Blumenthal |
| Canada First Research Excellence Fund | | Ali R Khan<br>Stefan Köhler |
| Brain Canada Platform Support Grant | | Ali R Khan<br>Stefan Köhler |

The funders had no role in study design, data collection, and interpretation, or the decision to submit the work for publication.

### Author contributions

Kayla M Ferko, Conceptualization, Data curation, Formal analysis, Investigation, Methodology, Project administration, Software, Visualization, Writing – original draft, Writing – review and editing; Anna Blumenthal, Conceptualization, Data curation, Investigation, Methodology, Writing – original draft, Writing – review and editing; Chris B Martin, Formal analysis, Writing – review and editing; Daria Proklova, Data curation, Formal analysis, Methodology, Software; Alexander N Minos, Data curation, Investigation; Lisa M Saksida, Timothy J Bussey, Conceptualization, Writing – review and editing; Ali R Khan, Data curation, Funding acquisition, Methodology, Software, Supervision; Stefan Köhler, Conceptualization, Formal analysis, Funding acquisition, Methodology, Project administration, Supervision, Writing – original draft, Writing – review and editing

### Author ORCIDs

Kayla M Ferko  http://orcid.org/0000-0003-4362-7295
Anna Blumenthal  http://orcid.org/0000-0002-8768-0189
Chris B Martin  http://orcid.org/0000-0002-7014-4371
Ali R Khan  http://orcid.org/0000-0002-0760-8647
Stefan Köhler  http://orcid.org/0000-0003-1905-6453

### Ethics

The study was approved by the Institutional Review Board at the University of Western Ontario (REB # 0442). Informed consent was obtained from each participant before the experiment, including consent to publish anonymized results.

### Decision letter and Author response

Decision letter https://doi.org/10.7554/eLife.66884.sa1
Author response https://doi.org/10.7554/eLife.66884.sa2

## Additional files

### Supplementary files

• Supplementary file 1. Behavioural performance on Category-Exemplar 1-Back Task. Proportion of correct responses for each trial type are indicated in green.
• Supplementary file 2. Multiple linear regression: brain RDM ~ (own RDM + average RDM).

• Transparent reporting form

### Data availability

All data generated or analyzed during this study are included in the manuscript and supporting fields. Source data files have been provided for Figures 1, 2, 3, 4, 6,7.

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
