## [Editor Report]

Your response has been thorough and thoughtful and we believe this work now represents an important advancement to our understanding of the contributions of anterior temporal lobe regions in visual representations. Your approach affords tremendous specificity in the conclusions one can draw about the relationship between visual similarity and neural similarity along this ventral visual pathway and highlights perirhinal cortex as a potential key region whose neural representational structure relates to subjective behavior.

---

## [Decision Letter]

**Decision letter after peer review:**

Thank you for submitting your article "Activity in perirhinal and entorhinal cortex predicts observer-specific perceived visual similarities between objects" for consideration by *eLife*. Your article has been reviewed by 2 peer reviewers, and the evaluation has been overseen by a Reviewing Editor and Christian Büchel as the Senior Editor. The following individual involved in review of your submission has agreed to reveal their identity: Simon W Davis (Reviewer #2).

Essential revisions:

Both reviewers agreed that the value of the results was clear, the methods are methodologically sound but that there are some theoretical or analytical ambiguities that need to be addressed.

The work is an important extension of prior work (particularly Charest et al., 2014), that includes adjacent brain areas, most notably the medial temporal lobe. However, the reviewers and I agreed that the theoretical basis and contributions of the results could use some further highlighting to underscore what new insights are gleaned from this work. In particular, we found the results in Figure 4 to be the most unique and important of this work but did not feel the Introduction set up the paper to appreciate this.

It was suggested that further analyses be conducted to boost the strength of the paper to resolve ambiguities about the subjective similarity measure to better understand what underlies these idiosyncratic representations (visual information? semantic information? familiarity?), and what these results suggest about the perirhinal cortex. Does this challenge or support any pre-existing notions about the perirhinal cortex, or the nature of visual / memory representations in the brain?

Specifically, please report representational dissimilarity matrices (RDMs) to identify what portions of the variance are idiosyncratic vs. shared across the group and attempt to relate the brain RDMs to visually-based RDMs of the stimuli. It was recognized that the semantic factor is much more constrained (given the use of a carefully controlled stimulus set) and is a strength of the current manuscript.

Other important potential additions for discussion are outlined in the individual reviewer comments.

*Reviewer #1 (Recommendations for the authors):*

Some of my struggle with the scope of the work may be in how it's framed in the introduction. The intro discusses the open question of how the brain represents similarities in objects, but then that question is never answered. It's also not clear to me what the key novelty is of the study beyond prior work, and how that changes our understanding of the human visual system. I'm also not sure why real-world objects are an interesting set or novelty (versus other types of stimuli we'd expect to have more idiosyncratic representations).

I also have the question of whether the offline object arrangement task could have influenced the similarity structure reflected in the brain.

Also, most of the analyses looked at correlations between individuals' brain and behavior to look at "perceived visual similarity". However, the behavioral measure includes both what we can consider the group-shared visual similarity (the mean, perhaps even capturing the "objective visual similarity"), and the individual-specific visual similarity (the variance). I am curious to see what regions still show a correlation with behavior even if you remove the mean -- so, for example, if you calculate an RDM factoring out / partial out the group mean, is it the PRC and alErC that show a correlation with really this individual-specific RDM? That being said, your later analysis showing higher correlations with an individual's behavioral RDM than other participants' RDMs answers this question in a different way. But the method I suggest also isolates the representation specific to the individual.

With that, I wonder if one can begin to look at the causes of these idiosyncracies. Since the current study is looking at visual similarity, I kept wondering if it would make sense to get an "objective" measure of visual similarity, by creating RDMs formed from some sort of computer vision metric (e.g., pixel similarity, or a DNN metric). You could examine the degree to which individuals agree / disagree with objective similarity, and how that relates to patterns in the brain. I wouldn't say this is necessary, but one potential direction that could expand the impact of the current work.

*Reviewer #2 (Recommendations for the authors):*

The authors argue in the Discussion for a modest relationship between perceived and objective measures of similarity (p23), and as such "the coding of whole objects based on complex feature conjunctions in PrC and alErC could afford the flexibility that is required to capture these differences in perception". What does "complex" mean here? Specifically, it is unclear as to whether the authors are suggesting that "subjectively perceived aspects" of stimuli constitute just another set of features (in addition to objective perceptual/semantic features), or something else entirely.

One inference the paper makes is that all 10 items contribute to the PRC/ERC effects equally. Do the authors have any evidence for this? How similar were the 10 different item categories? More specifically, did they have similar between-exemplar similarity ratings? This could maybe be inferred maybe from SFig1B for 5 of the items.

Absent from much of the framing in this paper is the characterization of perceptual verses semantic similarity, in favor of a coarse/fine-grained motif. However, some of these elements may be influencing the Supplementary analysis of centroid similarity. For example, in the first stage of the iMDS when all 10 categories of items are on the screen, it's unlikely that raters can do a purely visual decomposition of the visual similarity, despite the instructions. As such, there may be some "semantic similarity" in these groupings (and the consequent centroids). Could the authors speak to whether their Supplementary analyses of these relationships (SFig4) address this semantic factor?

Do the authors see any similarities in their LOC result for SFig4B, and the ITC-centered results in Charest et al., 2014? Some mention of this in the Discussion paragraph might help to draw further links between these papers.

---

## [Author Response]

Essential revisions:Both reviewers agreed that the value of the results was clear, the methods are methodologically sound but that there are some theoretical or analytical ambiguities that need to be addressed.The work is an important extension of prior work (particularly Charest et al., 2014), that includes adjacent brain areas, most notably the medial temporal lobe. However, the reviewers and I agreed that the theoretical basis and contributions of the results could use some further highlighting to underscore what new insights are gleaned from this work. In particular, we found the results in Figure 4 to be the most unique and important of this work but did not feel the Introduction set up the paper to appreciate this.

Response 1A: We thank the reviewers and editors for highlighting the methodological soundness of our work, and for spelling out what they consider the most important contribution in the presented set of findings. We share the excitement about the findings reported in Figure 4, which reveal that activity patterns in PrC and alErC predict observers’ perceived visual similarity structure among category exemplars with higher precision than more posterior VVS regions. We have introduced substantial revisions to the Introduction and Discussion in order to facilitate appreciation of these findings within the theoretical framework of the representational-hierarchical model of object processing in the VVS. In particular, we now elaborate on predictions as to how PrC and alErC compare with more posterior VVS regions (see pages 4 and 5). We elaborate on the corresponding interpretation of the reported results, while highlighting our new contribution, on p 29, 31, and 32. We also make reference to the inclusion of a new comparison between subjective ratings of perceived similarity and objective image characterization based on computational modeling (for further detail, see Essential Revisions Response 1B, 2B). The added findings we report based on this modeling, as well as new findings obtained with ratings averaged across observers, provide further support for the representational-hierarchical model as discussed in the modified and expanded Discussion on p 33. We also consider on p 28 how our finding that representational structure related to fine-grained perceived similarity in PrC and alErC activity is predominantly observerspecific (idiosyncratic) can be accommodated within this model, when the well documented evidence for plasticity in these medial-temporal regions is brought into play.

Response 1B: In order to help interpret the results in Figure 4

(considered most exciting by the Reviewers) in a more theoretically focused manner, and relate them to the observer-specific effects, we conducted additional analyses that address the unique vs shared component of perceived similarity structure in relation to estimates derived from the computational HMAX model at different levels/grains of within-category similarity, as illustrated in the new Figure 6 and the new panel Figure 7C. The HMAX model is valuable in this context as it provides a means to estimate objective image-based visual similarity at the level of intermediate feature descriptions based on prior neurophysiological characterization of the VVS. These new analyses are presented in the new section “Patterns in Posterior VVS Regions Predict Similarity Structure that is Shared by Observers and Tied to Object Characteristics at the Intermediate Feature Level”, with additional expansion of the section “Patterns in PrC and alErC Predict Fine-Grained Perceived Visual Similarity Structure among Exemplars in an Observer-Specific Manner”.

We summarize the results in the following way: *”*Together, these results suggest that activity in VVS regions posterior to PrC and alErC capture the components of perceived visual similarity structure among exemplars that is shared by observers and that is closely related to object features at the intermediate feature level. At the same time, these neural representations in posterior VVS regions do not appear to allow for differentiation of exemplars at high levels of perceived similarity that tend to be observer-specific*.*”

Response 1C: We changed the Title of the paper to reflect our new emphasis to: “Activity in perirhinal and entorhinal cortex predicts perceived visual similarities among category exemplars with highest precision”. Moreover, we substantially revised the abstract accordingly.

It was suggested that further analyses be conducted to boost the strength of the paper to resolve ambiguities about the subjective similarity measure to better understand what underlies these idiosyncratic representations (visual information? semantic information? familiarity?), and what these results suggest about the perirhinal cortex. Does this challenge or support any pre-existing notions about the perirhinal cortex, or the nature of visual / memory representations in the brain?Specifically, please report representational dissimilarity matrices (RDMs) to identify what portions of the variance are idiosyncratic vs. shared across the group and attempt to relate the brain RDMs to visually-based RDMs of the stimuli. It was recognized that the semantic factor is much more constrained (given the use of a carefully controlled stimulus set) and is a strength of the current manuscript.

Response 2A: Idiosyncratic vs Shared Variance & Computational Model-derived objective visual similarity. To better understand idiosyncratic versus shared aspects of ratings of perceived similarity, we introduced new analyses with an average perceived similarity RDM using the Means of ratings on the iMDS across all participants. We calculated brain-behaviour correlations using these average perceived similarity ratings (page 21-22; new Figure 6A-B p 23), and now present these in addition to the previously included analyses that focus on idiosyncratic aspects, which we expanded to allow for an examination of idiosyncratic structure at different grains (based on i-index as shown in Figure 7A-B). Our analyses revealed that average similarity ratings are not reflected in the representational structure of activity patterns in PrC and alErC but rather in EVC and LOC, and only for low and medium levels of similarity. We also related the brain RDMs to RDMs derived from the influential computational HMAX model that describes visual objects at the intermediate feature level (Riesenhuber, Poggio, 1999; Cadieu et al., 2007; Serre, Olivia, Poggio, 2007; Khaligh-Razavi, Kriegeskorte, 2014; see Figure 6C-D; see Methods section on page 37). Critically, we found that average perceived similarity that is shared across observers is tied to objective image characteristics at the intermediate feature level as estimated by the HMAX model, again only for low and medium levels of similarity though (see new behavioural results on p 8 and new neuroimaging findings presented in Figure 7C-D). At the most fine-grained level of similarity (i.e. among highly similar objects), neither average ratings nor HMAX estimates predicted activity patterns in any VVS region we examined. This pattern of results underscores our point that the fine-grained structure in perceived similarity among category exemplars that is captured by PrC and alErC representations is predominantly idiosyncratic, as supported by new behavioural analyses that focused on intersubject correlations in perceived similarity at different grains (see p 8).

Response 2B: Computational Model-derived objective visual similarity. We note that we also explored use of a deep convolutional neural network, including AlexNet and VGG-16. However, this approach was not well suited given our stimuli are in greyscale format, leading to limited classification performance in output. For example, apple was classified as “golfball”. In addition, we browsed various discussion boards (e.g., https://stackoverflow.com/questions/44668771/can-we-use-the-weights-of-a-model-trained-onrgb-images-for-grayscale-images; https://datascience.stackexchange.com/questions/22684/is-itpossible-to-use-grayscale-images-to-existing-model) and explored using a neural network that was trained using greyscale ImageNet images (https://github.com/DaveRichmond-/grayscaleimagenet/blob/master/eval_image_classifier_gray.py). The latter has potential to classify categories correctly, but the network architecture becomes arguably less ecologically valid. Consequently, we limit presentation of computational modeling results to those obtained with HMAX in the current paper, which still help constrain interpretation of our main findings, as discussed above.

Response 2C: Theoretical interpretation. Please see our extended Discussion section on page 31, 32 for theoretical interpretation of these sets of new results in the context of the representational-hierarchical model of VVS organization and the functional role of PrC and alErC. We argue that our pattern of findings provides evidence in support of a central claim of the representational-hierarchical model, namely that PrC computes more integrated object representations allowing for increased differentiation of objects in visual perception than more posterior VVS regions that code for objects at the intermediate feature level. We also suggest that idiosyncratic fine-grained similarity structure among exemplars in PrC/alErC, perceptual reports, and discrimination behaviour could reflect interindividual differences in perceptual expertise, an account that receives some indirect preliminary support from findings reported in the recent behavioural literature. We discuss this idea and indicate that it can directly be tested in follow-up fMRI research (p 33).

Other important potential additions for discussion are outlined in the individual reviewer comments.Reviewer #1 (Recommendations for the authors):Some of my struggle with the scope of the work may be in how it's framed in the introduction. The intro discusses the open question of how the brain represents similarities in objects, but then that question is never answered. It's also not clear to me what the key novelty is of the study beyond prior work, and how that changes our understanding of the human visual system. I'm also not sure why real-world objects are an interesting set or novelty (versus other types of stimuli we'd expect to have more idiosyncratic representations).

Please see Essential Revisions – Response 1A and 2C and Reviewer 1 – Response 2 in the Public Review.

I also have the question of whether the offline object arrangement task could have influenced the similarity structure reflected in the brain.

Please see Reviewer 1 – Response 1 in the Public Review.

Also, most of the analyses looked at correlations between individuals' brain and behavior to look at "perceived visual similarity". However, the behavioral measure includes both what we can consider the group-shared visual similarity (the mean, perhaps even capturing the "objective visual similarity"), and the individual-specific visual similarity (the variance). I am curious to see what regions still show a correlation with behavior even if you remove the mean -- so, for example, if you calculate an RDM factoring out / partial out the group mean, is it the PRC and alErC that show a correlation with really this individual-specific RDM? That being said, your later analysis showing higher correlations with an individual's behavioral RDM than other participants' RDMs answers this question in a different way. But the method I suggest also isolates the representation specific to the individual.

Reviewer 1 – Response 4: In addition to the results derived from the brain-behaviour i-index, we have included supplementary supporting analyses using multiple linear regression of the kind the Reviewer suggests in order to determine the contributions of shared versus observer-specific perceived similarity structure to activity patterns in our ROIs (p 23 main text; supplementary table 2). The results of these analyses for observer-specific effects closely parallel those reported for the i-index and converge on the same conclusions.

With that, I wonder if one can begin to look at the causes of these idiosyncracies. Since the current study is looking at visual similarity, I kept wondering if it would make sense to get an "objective" measure of visual similarity, by creating RDMs formed from some sort of computer vision metric (e.g., pixel similarity, or a DNN metric). You could examine the degree to which individuals agree / disagree with objective similarity, and how that relates to patterns in the brain. I wouldn't say this is necessary, but one potential direction that could expand the impact of the current work.

Reviewer 1 – Response 5: We thank the Reviewer for this insightful comment and promising suggestion. We appreciate the value of contrasting the perceived visual similarity measures with objective ones. Please see Essential Revision – Response 2 above for a summary of the newly added analyses that focused on average similarity (across observers) and HMAX-derived objective similarity, as suggested. By revealing a different relationship to posterior VVS regions versus PrC and alErC, they offer further constraints for interpretation of the transformation of representations between these regions.

Reviewer #2 (Recommendations for the authors):The authors argue in the Discussion for a modest relationship between perceived and objective measures of similarity (p23), and as such "the coding of whole objects based on complex feature conjunctions in PrC and alErC could afford the flexibility that is required to capture these differences in perception". What does "complex" mean here? Specifically, it is unclear as to whether the authors are suggesting that "subjectively perceived aspects" of stimuli constitute just another set of features (in addition to objective perceptual/semantic features), or something else entirely.

Reviewer 2 – Response 2: We thank the reviewer for this comment – this is a great question. In the representational hierarchical model of VVS organization that guided our work “Complex” refers to the degree of integration based on feature conjunctions as we now discuss in more detail in the Introduction and Discussion. Because similar exemplars of the same category share many features, differentiation among them is possible by way of representing exemplars as a conjunction of many of their features. For example, in order to compare two apples and discriminate them from one another, it may not be sufficient to consider shape or shading in isolation, but necessary to represent differences in shape and shading in a conjoint manner. Several prior fMRI studies and studies in rodents that we cite in the Introduction provide evidence in support of the view that disambiguation of feature overlap is a critical factor in the computation of representations in PrC, and that feature conjunctions play a critical role in this disambiguation. Recent fMRI work also suggests that this applies to perceptual as well as semantic features (Martin et al., Neuron, 2018). The novelty of the approach we took here to test predictions of the representational hierarchical model for differentiation of category exemplars along the VVS, however, was to rely on subjective reports of perceived similarity that could be quantified on a single perceptual dimension rather than based on objective feature statistics. In as much as participants were explicitly asked to provide their ratings of similarity based on visual characteristics in the iMDS task, we argue that they provide a measure of visual similarity and that the corresponding neural representational structure also taps into organization based on visual characteristics. Indeed our newly added results from computational modeling with HMAX provide support for this perspective as they indicate that the reported perceived similarity and related activation patterns in the VVS are correlated with similarity estimates based on objective image characteristics in the visual domain (see Essential Revisions – Response 1B, 2A). By extension, we also suggest in the Discussion (p 33) that the variability in fine-grained representational structure across observers we report for PrC and ErC could also reflect variability in visual representation, specifically variability based on expertise. Here we raise the possibility that it reflects differences in perceptual expertise rather than semantic knowledge and we make reference to pertinent behavioural research in the literature. This interpretation is admittedly speculative, and we present it as an idea that deserves to be tested directly in future fMRI research with training paradigms (see also Reviewer 1 – Response 2).

One inference the paper makes is that all 10 items contribute to the PRC/ERC effects equally. Do the authors have any evidence for this? How similar were the 10 different item categories? More specifically, did they have similar between-exemplar similarity ratings? This could maybe be inferred maybe from SFig1B for 5 of the items.

Reviewer 2 – Response 3: This is an interesting question to ask when considering that we used multiple categories, and that the main analyses focused on examination of structure in matrices that included data for all categories combined. We explored this question in further behavioural analyses that examined each category in isolation, but we did not find noticeable category differences. Figure 1-Supplementary Figure 1B now includes behavioural data for all 10 categories. Given that we focused on a relatively large number of categories with a limited number of exemplars in each of them, however, it is possible that our design did not have sufficient power to uncover category-specific effects. As such we feel the absence of category specific effects requires caution in interpretation, and we decided against bringing them into the main text and against including them in our analyses of similarity structure in different VVS regions.

Absent from much of the framing in this paper is the characterization of perceptual verses semantic similarity, in favor of a coarse/fine-grained motif. However, some of these elements may be influencing the Supplementary analysis of centroid similarity. For example, in the first stage of the iMDS when all 10 categories of items are on the screen, it's unlikely that raters can do a purely visual decomposition of the visual similarity, despite the instructions. As such, there may be some "semantic similarity" in these groupings (and the consequent centroids). Could the authors speak to whether their Supplementary analyses of these relationships (SFig4) address this semantic factor?

Reviewer 2 – Response 4: We appreciate the note of caution expressed by the Reviewer with respect to interpretation of the data for the first stage of the iMDS that required judgment of similarity among exemplars across all categories examined. We agree that the data for this component of the task, which were only used for characterization of category centroids and neural correlates in Supplementary material, may not provide an estimate of similarity that is exclusively tied to visual characteristics (despite instructions asking it), but may be contaminated by semantic factors that become more prominent in categorization as compared to subordinate exemplar discrimination. With this concern in mind, we removed any reference to centroid representations and data from the first phase of the iMDS task in our paper and Supplementary materials. Further, we now spell out more clearly in numerous places, including in the changed Title of our paper, that this study specifically focuses on similarities among (subordinate) category exemplars. For further detail on our perspective on visual vs semantic similarities among exemplars, as probed in our study, please see Reviewer 2 – Response 2.

Do the authors see any similarities in their LOC result for SFig4B, and the ITC-centered results in Charest et al., 2014? Some mention of this in the Discussion paragraph might help to draw further links between these papers.

Reviewer 2 – Response 5: Thank you for this helpful suggestion. We added direct comparison of results in IT and LOC for a more complete documentation of VVS findings in Figure 7Supplementary Figure 1. IT is a large swath of cortex that includes ROIs LOC and PhC in the current study. The stimuli used in our study were objects that varied in shape (as well as texture), and therefore we were specifically interested in probing LOC. PhC was used as a control region for comparison and we did not predict any significant correlations with perceived visual similarity of objects in this region, given its well established role in scene processing. The results across four core sets of analyses were highly similar between IT and LOC, however, with the only noticeable exception being that LOC activity patterns predict observer-specific perceived similarity at low levels of similarity, while IT patterns do not predict any observer-specific similarity at all.